# *Ocotea glomerata* (Nees) Mez Extract and Fractions: Chemical Characterization, Anti-*Candida* Activity and Related Mechanism of Action

**DOI:** 10.3390/antibiotics9070394

**Published:** 2020-07-09

**Authors:** Mayara Nunes Vitor Anjos, Luiz Nascimento de Araújo-Neto, Maria Daniela Silva Buonafina, Rejane Pereira Neves, Edson Rubhens de Souza, Isabelle Cristinne Ferraz Bezerra, Magda Rhayanny Assunção Ferreira, Luiz Alberto Lira Soares, Henrique Douglas Melo Coutinho, Natália Martins, Márcia Vanusa da Silva, Maria Tereza dos Santos Correia

**Affiliations:** 1Laboratory of Natural Products, Department of Biochemistry, Federal University of Pernambuco, Recife 50670-901, Brazil; mayanjos@hotmail.com (M.N.V.A.); marcia.vanusa@ufpe.br (M.V.d.S.); mtscorreia@gmail.com (M.T.d.S.C.); 2Laboratory of Medical Mycology, Department of Mycology, Federal University of Pernambuco, Recife 50670-901, Brazil; luizneto.neo@gmail.com (L.N.d.A.-N.); danielabuonafina@hotmail.com (M.D.S.B.); rejadel@yahoo.com.br (R.P.N.); edsonrubhens@hotmail.com (E.R.d.S.); 3Laboratory of Pharmacognosy, Department of Pharmaceutical Sciences, Federal University of Pernambuco, Recife 50670-901, Brazil; isabelle.ferraz@outlook.com (I.C.F.B.); magda.ferreira00@gmail.com (M.R.A.F.); phtech@uol.com.br (L.A.L.S.); 4Laboratory of Microbiology and Molecular Biology, Department of Biological Chemistry, Regional University of Cariri, Crato 63000-000, Brazil; 5Faculty of Medicine, University of Porto, Alameda Prof. Hernâni Monteiro, 4200-319 Porto, Portugal; 6Institute for research and Innovation in Health (i3S), University of Porto, Rua Alfredo Allen, 4200-135 Porto, Portugal

**Keywords:** natural products, *Ocotea glomerata*, anti-*candida* activity, mechanism of action

## Abstract

Background: Opportunistic fungal infections are increasingly common, with *Candida albicans* being the most common etiological agent; however, in recent years, episodes of candidiasis caused by non-*albicans*
*Candida* species have emerged. Plants belonging to the Lauraceae family have shown remarkable antifungal effects. This study assessed the anti-*Candida* activity of *Ocotea glomerata* extracts and fractions, time of death and the synergistic effects with conventional antifungals. The possible mechanism of action was also addressed. Methods: Minimal inhibitory concentrations (MIC) were determined by broth microdilution technique, and the mechanism of action was assessed by ergosterol, sorbitol, cell viability, reactive oxygen species (ROS) generation and phosphatidylserine externalization tests. Results: All the tested extracts evidenced antifungal activity, but the methanol extract was revealed to be the most effective (MIC = 3.12 μg/mL) on *C. krusei*. The combination of methanol extract with ketoconazole and fluconazole revealed a synergistic effect for *C. krusei* and *C. albicans*, respectively. Fractions 1 and 5 obtained from the methanol extract had fungicidal activity, mainly against *C. krusei*. Methanol extract did not reveal effects by ergosterol and sorbitol assays; however, it led to an increase in intracellular ROS levels, decreased cell viability, and consequently, cell death. Conclusion: *O. glomerata* methanol extract may be viewed as a rich source of biomolecules with antifungal activity against *Candida* spp.

## 1. Introduction

*Candida* species has a high adaptive potential, being able to develop in diverse nutrient-rich environments, and under distinct temperature, pH, osmolarity and oxygen quantity conditions [1]. The geographical distribution of *Candida* spp. has changed over the past few decades, with a decrease in the incidence of *C. albicans* and an increase in cases of non-*albicans Candida* species. Although geographical distribution has a high impact, the clinical features of patients also pose a trigger. The epidemiological modification of *C. albicans* infections in relation to non-*albicans Candida* species is clinically relevant due to the high rates of resistance to fluconazole among some species [2]. The high mortality rates, mainly in the Intensive Care Units (ICU), correspond to 15% of the cases of candidemia and are responsible for 50–70% of the systemic cases of fungal infections [3,4]. These characteristics associated with a high resistance to antifungals, virulence features and its ability to form biofilms with other species [5], turn *Candida* spp. into a threat to human health. There are at least 15 distinct *Candida* species that cause human disease, but more than 90% of invasive diseases are caused by the 5 most common species, namely *C. albicans*, *C. glabrata*, *C. tropicalis*, *C. parapsilosis* and *C. krusei*. Each of these organisms has a unique virulence potential, susceptibility to antifungals and epidemiology [6]. In addition, consistent increases in resistance to traditional antifungals have resulted in the need to control *Candida* spp. infections by early diagnosis and candidiasis prevention [3]. Thus, the discovery of effective antifungals with low toxicity, broad-spectrum activity and new mechanisms of action has become ever more important [7,8].

Natural products provide the basis for many anti-infectious therapies currently in use, including polyenes, echinocandins and flavonoids, three major classes of naturally antifungal drugs [9,10]. With this in mind, the Lauraceae botanical family, known for its commercial interest owing to its essential oils, includes approximately 50 genera and 2500 species, of which the *Ocotea* genus stands out [11]. Many reports in the literature have addressed the chemical composition of *Ocotea* oils that occur worldwide [12,13]. Studies on the biological properties of *Ocotea* essential oil reveal a broad-spectrum activity, including antimicrobial effects [14,15]. However, the evaluation of the antifungal activity of extracts and fractions, especially its anti-*Candida* activity, obtained from *Ocotea glomerata* (Nees) Mez, has not yet been elucidated. Thus, the present study aimed to chemically characterize the extracts of *O. glomerata,* as well as to determine its antifungal activity, mainly against *Candida* spp., and to elucidate its mechanism of action.

## 2. Results and Discussion

### 2.1. Identification of the Ocotea glomerata Methanol Extract and the Fraction Components

Chromatograms obtained from *O. glomerata* methanolic extract (OGME) and fractions using the scanning spectra corresponding to the major peaks at 280 and 350 nm are shown in Table 1. For OGME, the presence of cinnamic derivatives and flavonoids (25.377; 27.223 min) were observed. For methanol fraction 1 (FEM1), the presence of gallic acid was observed at 6.443 min, along with others cinnamic derivatives (5.650 min) and flavonoids (25.293 min), which correspond to that found in the OGME, indicating extract fractionation. In FEM2 and FEM3, cinnamic acid derivatives and flavonoids were observed, but in FEM4 and FEM5, only flavonoids were identified. The flavonoids found in FEM5 (25.263 and 27.177 min) correspond to the flavonoids in OGME.

### 2.2. Antifungal Susceptibility Testing

In the search for new phytotherapics, flavonoids present in medicinal plants are strong therapeutic agents in cases of microbial disease, in addition to having antiviral, antiproliferative, antimutagenic, antithrombotic and antioxidant activities [16]. In this sense, the presence of phenolic compounds in the Lauraceae family characterizes a promising new model for the evaluation of antimicrobial mechanisms, with their effects resulting from interactions of compounds in the cell membrane microorganisms, probably due to their ability to complex with intracellular proteins and cell wall [17].

The results of the antifungal activity of OGME and its fractions are shown in Table 2 and were performed according to the standards established by the Clinical and Laboratory Standards Institute [18] using the M27-S4 protocol. OGME exhibited an antifungal activity profile ranging from 3.12 to 400 µg/mL for all strains tested, with the exception of *C. tropicalis* URM6551, which was sensitive only to the hexane extract (OGHE; 100 µg/mL). The chloroform (OGCE), OGHE and the ethyl acetate (OGAcOEt) extracts presented inhibitory concentrations ranging from 100 to 400 µg/mL. However, at concentrations tested, *C. albicans* ATCC14053 was not sensitive to the extracts.

OGME was the most promising, presenting the lowest minimal inhibitory concentration (MIC) for *C. krusei* ATCC 6258 (MIC = 3.12 µg/mL). This value was almost equal to that of ketoconazole (MIC = 2.0 µg/mL). However, the proven hepatotoxicity of azolics makes its moderate use difficult, as treatment with this drug requires that the serum concentration remains at higher levels in the body, above the MIC, in most dosage intervals. Thus, adverse reactions can range from mild transient elevations in transaminase levels to hepatitis, cholestasis and fulminant hepatic failure [19]. In this sense, natural products can be a safer and effective alternative for the treatment of fungal infections due to their low associated toxicity [20]. *C. albicans* URM5901, *C. parapsilosis* ATCC22019 and URM7048 presented MIC values of 6.25 µg/mL. These data reveal antifungal potential for *Candida* infections, once a dose-dependent response for fluconazole (16.0 µg/mL) was found for *C. albicans* URM5901, more sensitive to OGME. For *C. parapsilosis* URM4970, a MIC value of 12.5 μg/mL was found. However, when compared to fluconazole (8.0 μg/mL), this extract may be viewed as promising, as fluconazole only exhibits a fungistatic action while the tested compounds exert a fungicidal action [21]. For the *C. krusei* ATCC6258 strain, a MIC value of 3.12 µg/mL was obtained to OGME. This species has emerged in recent years as an opportunistic fungi with a high degree of complexity in infected patients, especially in those who are immunocompromised. The intrinsic resistance to fluconazole and the higher rates of amphotericin B inefficacy against *C. krusei* infection is a cause for concern. In this sense, the search for new drug approaches able to inhibit *Candida* species’ growth and infection is extremely relevant [22].

In view of the obtained MIC values, OGME displayed the greater anti-*Candida* potential. This extract was fractionated with a H_2_O:MeOH (90%:20%) mobile phase using flash chromatography, where 5 fractions were separated by their spectra (peak absorption range) represented by specific staining. Separation efficiency was associated with particle and column size. The antifungal activity of FEM1 to FEM5 are also shown in Table 2. The FEM1 fraction did not reveal a MIC for *C. albicans* at the tested concentrations; however, a MIC of 400.0 μg/mL was found for *C. parapsilosis*. In contrast, for *C. tropicalis* URM6551 and *C. krusei* ATCC6258, MICs of 200.0 μg/mL were obtained. In general, the FEM5 fraction was more effective when compared to other fractions, where MIC values of 400.0 μg/mL were obtained for all *C. albicans* and *C. parapsilosis* URM4970 strains, 200.0 μg/mL for *C. tropicalis*, 50.0 μg/mL for *C. parapsilosis* URM7048 and 25.0 μg/mL for *C. kusei* ATCC6258. FEM2, FEM3 and FEM4 fractions did not exhibit antifungal activity at the tested concentrations. Results obtained for fractions revealed a lower anti-*Candida* activity when compared to OGME. This finding may be explained by the fact that possibly, compounds present in the extract act synergistically, while in fractions, they are separated, thus reducing the fractions’ potential.

The reading of the results does not present a consensual classification in relation to MIC values obtained by natural products [23]. Thus, considering MIC values equal to or below 500 μg/mL as strong inhibitors, MIC values between 600 and 1500 μg/mL as moderate inhibitors, MIC values above 1600 μg/mL as weak inhibitors and MIC values equal to or below 1000 μg/mL as satisfactory [24], our results are promising, as they are below those established by these authors.

### 2.3. Checkerboard Analysis

In addition to its inherent antimicrobial properties, natural products and their derivatives may alter the effects of standard antifungal agents. The combination of two or more antifungal agents may lead to a reduction in the required drug dosages and to a decrease in adverse events. The in vitro assessment of synergistic, additive or antagonistic interactions between antifungal drugs against microbial strains can be performed using the “checkerboard” test [25].

OGME was used in the checkerboard analysis and combination results are shown in Table 3. Additive effects were observed for extract combinations with ketoconazole and fluconazole for *C. parapsilosis* ATCC22019, and extract combinations with ketoconazole for *C. albicans* ATCC14053; moreover, extract combinations with ketoconazole and fluconazole evidenced a synergistic effect for *C. krusei* ATCC6258 and *C. abicans* ATCC14053 strains. Extract combinations with amphotericin B revealed an indifferent effect against the tested strains, where OGME did not potentiate nor antagonize the amphotericin B effect. As OGME did not present any type of association with amphotericin B, this finding reinforces the hypothesis that the active compound present in the extract has a distinct mechanism of action from that of the antifungal agent against *Candida* spp.

According to the data shown in Table 3, OGME used in combination with antifungal agents resulted in a MIC reduction for ketoconazole against all *Candida* strains tested, and for amphotericin B, the MIC value also decreased from 0.06 to 0.03 μg/mL against *C. krusei* ATCC6258. In addition, the MIC of fluconazole decreased from 1.0 to 0.06 μg/mL and from 16.0 to 0.12 μg/mL against *C. parapsilosis* ATCC22019 and *C. albicans* ATCC14053, respectively. OGME positively modulated the in vitro action of antifungals and their combination led to synergistic effects, since an increase in yeasts death at lower antifungal concentrations occurred, suggesting their potential use as adjuvant in the anti-*Candida* treatment. 

Several studies have been performed using different antifungal combinations with plant extracts, however, the OGME combination with synthetic drugs against *Candida* strains is reported here for the first time.

### 2.4. Sorbitol and Ergosterol Test

The evaluation of a new drug with action on the cell wall can be verified through a sorbitol test. The test compares the MIC of the antifungal drug in the presence and absence of sorbitol (osmotic shield used to stabilize fungal protoplasts). If a drug acts in any way on the fungal cell wall, it triggers cell wall lysis when in the absence of an osmotic stabilizer; however, it will allow its growth in the presence of this osmotic support [26]. In this study, sorbitol tests showed that OGME has no antifungal effect on *C. krusei* ATCC6258, as the MIC values obtained in the presence and absence of sorbitol were the same, suggesting that the extract acts on another cell target. 

Due to the importance of ergosterol in maintaining cell growth and in the function of fungal membranes, this sterol can be used as a target for the development of new drugs. Amphotericin B can be used as a positive control, given its interaction with ergosterol [27]. In this sense, an ergosterol test was carried out using a method based on the exogenous supply of sterol to a compound that, with affinity for sterols, will quickly form a complex, protecting membrane sterols. This results in an increase in MIC [28]. In this way, the MIC values of OGME and its active fractions FEM1 and FEM5 were assessed on *C. krusei* ATCC6258 strain, in the absence and presence of ergosterol. No changes in MIC values were stated in media with and without the addition of ergosterol, indicating that they have no action on ergosterol. However, the MIC of amphotericin B had its value increased in the presence of exogenous ergosterol, proving to be a positive control.

### 2.5. Cell Viability

Data obtained for cell viability analysis using the Propidium Iodide (PI) marker are shown in Figure 1. *C. krusei* cells, when treated with OGME at different concentrations (MIC, 2MIC, 4MIC) 1.35 × 10^6^ ± 0.04 cells, 0.64 × 10^6^ ± 0.04 cells and 0.30 × 10^6^ ± 0.01 cells respectively, showed a significant decrease (*p* < 0.05) in cell viability, in a dose-dependent manner. Amphotericin B, used as a positive control, led to a decrease in viable cells number (0.28 × 10^6^ ± 0.02 cells) compared to the negative control, 24 h after exposure. Thus, our data corroborate the findings obtained by Da Silva et al. [29], who also observed a dose-dependent decrease in cell number, when comparing 3 naphthoquinone compounds against fluconazole-resistant *C. tropicalis* cells. PI is a fluorescent dye that interleaves with the DNA and RNA of cells whose plasma membrane integrity has been lost [20]. Therefore, the labeling of cells with this compound allows an analysis of the antifungal activity of OGME in biological membranes, as occurs with the action of amphotericin B [30].

### 2.6. Detection of Phosphatidylserine (PS) in C. krusei Cells Induced by Methanol Extract

Phosphatidylserine (PS) release is an early event of cells entering apoptosis. In yeasts, PS is predominant in the internal monolayer of the lipid bilayer of the cytoplasmic membrane and is translocated to the outer monolayer during apoptosis [31]. Annexin V, a phospholipid binding protein with high affinity due to polarity loss, capable of binding to the released PS, was used for PS detection [32]. Figure 2 shows the results obtained for OGME treatments against *C. krusei* marked with annexin V. After 24 h of exposure, the OGME, at different concentrations (MIC, 2MIC and 4MIC), led to a significant increase (*p* < 0.05) in the frequency of *C. krusei* cells releasing PS, when compared to the control (1.25 ± 0.03%). At MIC, 2MIC and 4MIC values, 45.73 ± 2.33%, 63.29 ± 1.72% and 75.57 ± 1.30% of cells with PS were identified, respectively. The positive control, amphotericin B, led to a percentage of cells with PS of 64.02 ± 1.62%.

After exposure to OGME at different concentrations, *C. krusei* ATCC6258 cells with PS conjugated with annexin V were observed in the first 3 h, confirming it as an initial effect on cells. This additional finding corroborates the activation pathways suggestive of apoptosis. Due to the loss of membrane polarity, PS is available in the external environment, stimulating the recognition and phagocytosis of the same cell by macrophages or other antigen-presenting cells [33].

### 2.7. Reactive Oxygen Species (ERO) Production

*C. krusei* ATCC6258 cells treated with OGME at different concentrations (MIC, 2MIC, 4MIC) presented a significant increase (*p* < 0.05) in intracellular ROS production in a dose-dependent manner (26.50 ± 0.59%, 44.53 ± 1.05%, 70.13 ± 1.39%, respectively), when compared to the control (1.48 ± 0.03%). OGME was able to induce ROS production at all tested concentrations in *C. krusei* cells, with values greater than that of amphotericin B, which was used as a positive control (Figure 3).

*C. krusei* cells exposed to OGME showed an increase in intracellular ROS levels production, proportional to the increase in the dose of the compound. These data are similar to that of Miao et al. [34], who demonstrated an increase in ROS levels in a dose- and exposure time-dependent manner for *C. albicans* isolates after treatment with shikonin. 

The literature reports that ROS are necessary and sufficient to induce yeast apoptosis [32,35], a fact observed in the first 3 h of exposure to OGME. Elevated levels of ROS can cause oxidative stress in yeast cells with subsequent formation of oxidized cell macromolecules, including lipids, proteins and DNA, triggering biochemical signals for initial apoptosis [31]. From the analyses carried out in this study, the biological prospecting of *O. glomerata* extracts was evidenced, mainly for the OGME, which showed antifungal activity against *Candida* spp. OGME was the most active extract against all the strains used, except for *C. tropicalis*. Additionally, with the fractionation of the methanol extract, lower MIC values were obtained when compared to the extract.

The checkerboard test showed synergism between OGME and the tested antifungals, indicating a positive modulation of the in vitro action of the antifungals, suggesting a future use as an adjuvant to these drugs. Also, OGME and its bioactive fractions did not present complexation mechanisms with ergosterol and sorbitol, possibly acting on other targets, such as with molecules that integrate the antioxidant system (e.g., ROS production with consequent oxidative stress and apoptosis signs).

## 3. Materials and Methods 

The study was conducted in accordance with the Basic and Clinical Pharmacology and Toxicology policy for experimental and clinical studies [36]. 

### 3.1. Extract Acquisition

The collection area is located in the municipality of Igarassu, 28 km from Recife, in the mesoregion of the Zona da Mata of Pernambuco, North coast of the state, between the coordinates 07°41′04.9″ and 07°54′41.6″ S; 34°54′17.6″ and 35°05′07.2″ W. Collections were performed from December 2014 to February 2015 and concentrated in a fragment identified as Piedade, with geographical coordinates of 7°49′12″ and 7°50′55″ S; 35°0′35″ and 34°59′21″ W, comprising an area of 305,787 ha. Samples of *O. glomerata* were collected and the botanical identification was carried out at the Herbarium of the Agronomic Research Institute of Pernambuco (IPA), Brazil, and the voucher specimen (IPA 90.944) was deposited in the herbarium (IPA). *O. glomerata* leaves were dried in an oven (36 °C) and subsequently macerated. The leaf powder (10 g) was mixed with 100 mL of an eluotropic series of hexane, chloroform, ethyl acetate and methanol. All samples were shaken until saturated (24 h). After 24 h, the samples were concentrated until dry in a rotary evaporator and stored at 4 °C for further analysis.

### 3.2. Methanol Extract Fractionation

The dried extract (150 mg) was fractionated by flash chromatography using Biotage Isolera One^®^ equipment with a Biotage^®^ SNAP KP-SIL C18 33 g column with a H_2_O:MeOH mobile phase and a mobile phase gradient of 5–20% MeOH (3 Column Volumes-CV), 20–30% MeOH (1CV), 30–30% MeOH (1.5CV) 30–40% MeOH (1CV), 40–40% MeOH (1.5CV), 40–50% MeOH (1CV), 50–50% MeOH (1.5CV), 50–100% MeOH (3CV), 100–100% MeOH (3CV). Flow (50 mL/min) was constant, with a maximum volume per fraction of 18 mL and a rack type of 16 × 150 mm.

### 3.3. Phytochemical Prospecting of Secondary Metabolites

Phytochemical prospection was performed according to Matos [37], in which extract fractions were placed on 5 × 10 cm chromatographic plates of silica gel (Macherey-Nagel). Plates were eluted with 10% methanol/chloroform (*v/v*) and methanyl/ethyl acetate (*v/v*) solutions, dried and developed in ultraviolet (UV) light.

### 3.4. Identification of Crude Methanolic Extract and Fraction Compounds by High-Performance Liquid Chromatography (HPLC)

Methanol extract samples were subjected to high-performance liquid chromatography (HPLC), in which 2 mg were weighed and diluted in 2 mL of water. The solution obtained was filtered through a PVDF (Polyvinylidene fluoride) membrane with a pore aperture equal to 0.45 μm. The analysis was performed using a chromatograph equipped with a photodiode array detector (wavelength 280 nm), 3.9 μm pre-column, 250 mm long column and 4.6 mm internal diameter, packed with silica chemically attached to the octadecylsilane group (5 μm), kept at room temperature (24 °C), and a mobile phase flow rate of 0.8 mL/min. The mobile phase consisted of water (trifluoroacetic acid 0.05%) as solvent A and methanol (trifluoroacetic acid 0.05%) as solvent B, both degassed in an ultrasonic bath and filtered through a membrane, with 0.45 μm pore. The injection volume used was 20 μL. Separation was performed using a linear gradient: 0–10 min, 10–25% B; 10–20 min, 25–40% B; 20–25 min, 40–75% B; 25–28 min, 75–10% B; 28–30 min, 75–10% B.

### 3.5. Antifungal Activity

*O. glomerata* antifungal activity was evaluated according to the CLSI M27-S4 [18] document standards. Strains of *C. albicans* (ATCC14053 and URM5901), *C. parapsilosis* (ATCC22019, URM4970, and URM7048), *C. krusei* ATCC6258 and *C. tropicalis* URM6551 were obtained from the American Type Culture Collection (ATCC) and from the URM Micoteca Collection from the Department of Mycology, Federal University of Pernambuco. The extracts were diluted in dimethyl sulfoxide (DMSO) at concentrations ranging from 800 to 3.125 µg/mL. The concentrations of antifungals followed the rules of the protocol. Furthermore, DMSO concentration did not exceed 1%. All tests were performed in triplicate with two independent occasions.

### 3.6. Checkerboard Analysis

The combination of the standard drugs, fluconazole, amphotericin B and ketoconazole with OGME was tested in triplicate against *C. albicans* ATCC14053, *C. krusei* ATCC6258 and *C. parapsilosis* ATCC22019 using the previously described method (CLSI M27-S4) [18]. The initial inoculum was prepared as described for the antifungal sensitivity test. The readings were visually determined and read at 24 and 48 h. Fractionated inhibitory concentration (FIC) was calculated for each combination to assess the antifungal interactions.

FIC indices were calculated as IFC^A^ + IFC^B^, where IFC^A^ and IFC^B^ represent the minimal concentrations that inhibit fungal growth for drug A and drug B, respectively: IFC^A^ = combined MIC^A^/isolated MIC^A^ and FIC^B^ = combined MIC^B^/isolated MIC^B^. An average FIC index was calculated based on the following equation: FIC index = FIC^A^ + FIC^B^. In addition, the interpretation was as follows: synergistic (<0.5), additive (0.5–1.0), indifferent (>1) or antagonistic (>4) [38].

### 3.7. Ergosterol Test

Ergosterol was prepared during the test procedure according to Leite et al. [39] with some modifications, with ergosterol being first triturated and dissolved in DMSO and 1% Tween 80. To verify if bonds were formed between compounds and sterol membranes, an experiment was performed according to the method described by Escalante et al. [28]. The fungal compounds and inoculum used were prepared according to the CLSI M27-S4 protocol [18]. The same procedure was performed for Amphotericin B, whose interaction with the ergosterol membrane is already known, thus serving as a positive control. Plates were incubated at 35 °C for 48 h. Results showing MIC values different from the antifungal activity were interpreted as positive.

### 3.8. Sorbitol Test

The assay was performed using RPMI 1640 (Roswell Park Memorial Institute) medium supplemented with and without sorbitol (control), using the broth microdilution method according to the CLSI M27-S4 protocol [18]. The 96-well plates were incubated at 35 °C for 48 h and results were read at 24 and 48 h. Results which showed MIC values different from the antifungal activity were interpreted as positive.

### 3.9. Cell Viability Determination

Strains were exposed to OGME (MIC, 2MIC and 4MIC), to the RPMI culture medium (negative control) and to amphotericin B (positive control, 4 μg/mL) for cell viability determination. Treated cells were incubated at 35 °C for 24 h. Subsequently, aliquots were collected at 3, 6, 12 and 24 h time intervals after exposure to the compounds. All experiments were performed in triplicate on 3 independent occasions. Cells were labeled with PI and analyzed by flow cytometry (Guava EasyCyte™ Mini System). For each experiment, 10,000 events were analyzed [29].

### 3.10. Phosphatidylserine Externalization Determination

Phosphatidylserine externalization detection was performed using fluorescent annexin V and PI markers. *C. krusei* ATCC6258 cells were collected by centrifugation and digested with Zymolyase 20T (2 mg/mL) (Seikagaku Corporation, Japan) in potassium phosphate buffer (PPB) (1M, pH 6.0) for 2 h at 30 °C, following prior exposure to OGME. Protoplasts were resuspended in a solution containing annexin V-FITC (fluorescein isothiocyanate) (Guava Nexin Kit, Guava Technologies, Inc., Hayward, CA, USA) and PI (Sigma, St. Louis, MO, USA) in the absence of light at 37 °C. After 20 min, suspensions were analyzed by flow cytometry (Guava EasyCyte™ Mini System). For each experiment, 10,000 events were evaluated [29,40].

### 3.11. Reactive Oxygen Species Detection

For intracellularly generated ROS, the probe diacetate 2’7’-dichlorodihydrofluorescein (H_2_DCFDA) was used. *C. krusei* ATCC6258 cells were incubated with H_2_DCFDA (20 μM) (Sigma, USA) for 30 min in the dark at 37 °C, following prior exposure to OGME. Then, cells were washed and resuspended in PBS (phosphate buffered saline) and analyzed by flow cytometry (Guava EasyCyte™ Mini System) [29,40].

### 3.12. Data Analysis

Data obtained were analyzed using the mean ± standard error of the mean (SEM) of three independent experiments. To verify the occurrence of significant differences between the distinct concentrations used, analysis of variance (ANOVA) followed by the Student-Newman-Keuls test (*p* < 0.05) was performed, using Prism program, v. 5.01 (GraphPad Software, San Diego, CA, USA).

## Figures and Tables

**Figure 1 antibiotics-09-00394-f001:**
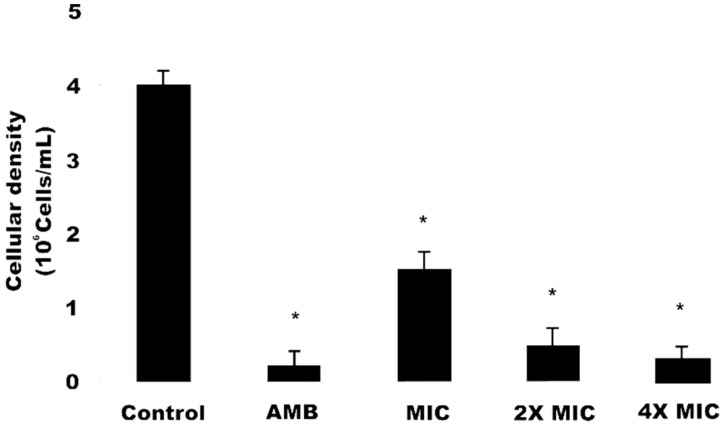
Effect of *O. glomerata* methanol extract on *C. krusei* cell viability using Propidium Iodide (PI). Data represent the mean and standard deviation (SD) of fluorescence during PI staining in two independent experiments with at least three replications. Analysis of variance (ANOVA) was performed followed by the Student-Newman-Keuls test (*p* < 0.05). * indicates significant differences between the fluorescence intensity percentage of the staining with PI (*p* ≤ 0.05). AMB, Amphotericin B; MIC, Minimum Inhibitory Concentration; 2MIC, Double the MIC value; 4MIC, Four times the MIC value.

**Figure 2 antibiotics-09-00394-f002:**
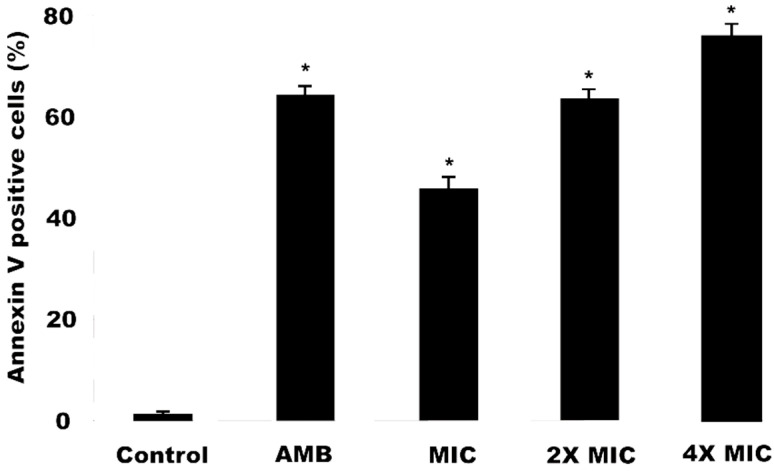
Effect of *O. glomerata* methanol extract on phosphatidylserine (PS) release in *C. krusei* cells. Data represent the mean and standard deviation (SD) of fluorescence during annexin-V labeling in two independent experiments with at least three replications. Analysis of variance (ANOVA) was performed followed by the Student-Newman-Keuls test (*p* < 0.05). * indicates significant differences between the percentage of fluorescence intensity of annexin-V label (*p* ≤ 0.05). AMB, Amphotericin B; MIC, Minimum Inhibitory Concentration; 2MIC, Double the MIC value; 4MIC, Four times the MIC value.

**Figure 3 antibiotics-09-00394-f003:**
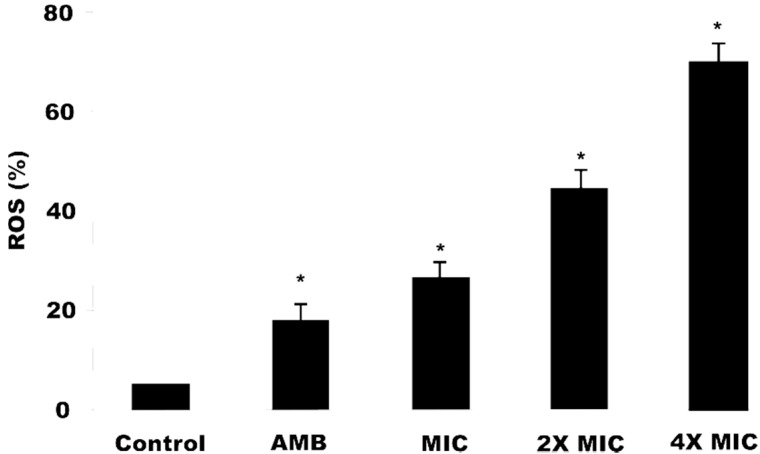
Reactive oxygen species (ROS) production in *C. krusei* cells after treatment with *O. glomerata* methanol extract. Data represent the mean and standard deviation (SD) of fluorescence during ROS production in two independent experiments with at least three replications. Analysis of variance (ANOVA) was performed followed by the Student-Newman-Keuls test (*p* < 0.05). * indicates significant differences between the percentage of fluorescence intensity of ROS production (*p* ≤ 0.05). AMB, Amphotericin B; MIC, Minimum Inhibitory Concentration; 2MIC, Double the MIC value; 4MIC, Four times the MIC value.

**Table 1 antibiotics-09-00394-t001:** *Ocotea glomerata* methanol extract and fraction components by High-Performance Liquid Chromatography (HPLC) (retention time in min).

	Cinnamic Derivatives			Flavonoids			
OGME	6.600			25.377;	27.223		
FEM1	5.650;	6.443		25.293			
FEM2	6.543;	9.860;	15.150	10.117;	25.263		
FEM3	10.070			25.263			
FEM4	-			25.267;	25.637;	26.127;	26.277
FEM5	-			25.263;	27.177		

OGME, *O. glomerata* methanolic extract; FEM1, Methanol fraction 1; FEM2, Methanol fraction 2; FEM3, Methanol fraction 3; FEM4, Methanol fraction 4; FEM5, Methanol fraction 5; -, non-existent.

**Table 2 antibiotics-09-00394-t002:** Minimal inhibitory concentration (MIC) of synthetic antifungals, extracts and active fractions of the *O. glomerata* methanol extract (μg/mL).

Species/Compounds	*C. albicans*		*C. parapsilosis*			*C. tropicalis*	*C. krusei*
	ATCC14053	URM5901	ATCC22019	URM4970	URM7048	URM6551	ATCC6258
OGHE	-	-	-	-	400	100	-
OGCE	-	400	200	100	100	-	100
OGAcOEt	-	200	-	400	100	-	100
OGME	400	6.25	6.25	12.5	6.25	-	3.12
FEM1	-	-	400	400	400	200	200
FEM2	-	-	-	-	-	-	-
FEM3	-	-	-	-	-	-	-
FEM4	-	-	-	-	-	-	-
FEM5	400	400	400	400	50	200	25
FLC	-	16	1	8	4	1	NA
KET	0.06	-	0.06	0.06	8	0.06	2
AMB	0.06	0.25	0.06	0.06	0.06	0.06	0.06

-, Not inhibitory; NA, Not analyzed; OGHE, hexane extract; OGCE, chloroform extract; OGAcOEt, ethyl acetate extracts; OGME, methanolic extract; FEM1, Methanol fraction 1; FEM2, Methanol fraction 2; FEM3, Methanol fraction 3; FEM4, Methanol fraction 4; FEM5, Methanol fraction 5; FLC, Fluconazole; KET, ketoconazole; AMB, Amphotericin B.

**Table 3 antibiotics-09-00394-t003:** MIC (μg/mL) and Fractionated Inhibitory Concentration (FIC) indices of antifungal drugs and the effect of their combination with *Ocotea glomerata* methanol extract.

Species/Compounds	*C. albicans* ATCC14053		*C. krusei* ATCC6258		*C. parapsilosis* ATCC22019	
	MIC	FIC Index	MIC	FIC Index	CIM	FIC Index
OGME	400		3.12		6.25	
AMB	0.06		0.06		0.06	
KET	0.06		2		0.06	
FLC	16		NA	NA	1	
OGME/AMB	12.5/0.06	1.03 (I)	1.56/0.03	1 (I)	3.12/0.06	1.5 (I)
OGME/ KET	6.25/0.03	0.51 (A)	0.78/0.03	0.26 (S)	0.78/0.03	0.62 (A)
OGME/FLC	6.25/0.12	0.02 (S)	NA	NA	3.12/0.06	0.56 (A)

ME, Methanol Extract; FLC, Fluconazole; KET, ketoconazole; AMB, Amphotericin B; NA, Not analyzed; FIC index, Interaction type (I, Indifferent; A, Additive and S, Synergic).

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
