# Peer review of "Ocotea glomerata* (Nees) Mez Extract and Fractions: Chemical Characterization, Anti-*Candida* Activity and Related Mechanism of Action"

_antibiotics, 2020, doi:10.3390/antibiotics9070394_

Round 1
Reviewer 1 Report
There is a new for new antifungal agents, especially those that can inhibit through novel mechanisms. The authors explore antifungal potential of Ocotea glomerate, a plant native to South America. Crude extracts are tested for inhibitory activity against Candida species. Initial evaluation of mechanistic action for inhibition suggest that ROS is built up in the fungal cells and could contribute to cell death. The manuscript would greatly benefit from knowing the compositional profile of the extracts or single active fraction.
Major:
- The manuscript needs assistance with English language. Additional language editing is strongly suggested.
- On line 47, the authors note that C. albicans pathogenicity is “scarcely” known. Actually, a great deal is known about C. albicans virulence and pathogenicity. There is still much to be learned but the current knowledge bank is not accurately described as “scarce”.
- Infections rates are described in the manuscript as increasing rapidly. This is not true for all parts of the world.
- In table 2, it is only “methanol” that exhibits any inhibitory activity. Is this unfractionated? Is this methanol alone as a negative control? It needs to be clarified what is in the “methanol” group.
- Line 93 list C. tropicalis URM6558 but the table list C. tropicalis URM6551.
- 400 mg/mL is a very high inhibitory concentration. Even considering that these are extracts where the active component is likely rather diluted.
- What is the compound composition within the selected fractions, especial FEM 5?
- A checkerboard assay is being done with the extract and fractions but there is no information provided on what the active element or compound composition of the extract. Seems like a premature assay when you don’t have the active component.
- What is the significance of the propidium iodide marker assay as it relates to identifying mechanism of action.
- You demonstrate a dose dependent increase in ROS with your provided methanol fraction that is not related to cell wall or ergosterol. One potential mechanism is the thioredoxin system that functions as a cellular antioxidant. To determine if this pathway is affected by your plant extracts, you can see if glutathione is an antagonist to the inhibitory activity.
Minor edits:
- In the abstract, line 25, remove the word “other”.
- Please revisit lines 38 and 39 from the abstract. The conclusion statement needs to be revised for English.
- Line 44: “many nutrients” should be changed to “diverse nutrient rich environments…”
- Lines 84-86, 99-101, 127, 137-138, 242-244 need editing.
Author Response
(1) The manuscript needs assistance with English language. Additional language editing is strongly suggested.
Answer: Dear reviewer, thank you for your comment. The text has been revised.
(2) On line 47, the authors note that C. albicans pathogenicity is “scarcely” known. Actually, a great deal is known about C. albicans virulence and pathogenicity. There is still much to be learned but the current knowledge bank is not accurately described as “scarce”
Answer: The text in the manuscript refers to the different species of Candida, and not just the species C. albicans. However, the text has been modified for a more general perspective regarding non-Candida albicans species.
(3) Infections rates are described in the manuscript as increasing rapidly. This is not true for all parts of the world.
Answer: The text has been modified. In fact, the growth of Candida infections depends on several factors, including local ones. Therefore, the text has been modified for more general information.
(4) In table 2, it is only “methanol” that exhibits any inhibitory activity. Is this unfractionated? Is this methanol alone as a negative control? It needs to be clarified what is in the “methanol” group.
Answer: Dear reviewer, thank you for your comment. The text has been modified so that the codes used in Table 2 and Table 1 correspond to each other. There was an error in the codes, but it has been fixed. The "methanol" refers to the crude methanolic extract of ocotea glomerata (OGME).
(5) Line 93 list C. tropicalis URM6558 but the table list C. tropicalis URM6551.
Answer: The registration number has been corrected in the text. The correct one is URM6551.
(6) 400 mg/mL is a very high inhibitory concentration. Even considering that these are extracts where the active component is likely rather diluted.
Answer: MIC values equal to 400 μg/mL are considered as strong inhibitors, according to the consensual classification in relation to MIC values obtained by natural products mentioned in the text. Reference was made to the study by Webster and collaborators (https://doi.org/10.1016/j.jep.2007.09.014). The MIC values are in micrograms per milliliter and not in milligrams per milliliter.
(7) What is the compound composition within the selected fractions, especial FEM 5?
Answer: We attributed to the flavonoids the antifungal activity observed in the methanolic fraction 5 as seen in tables 01 and 02. However, it was not possible to isolate and identify the chemical structure of the compound due to the lack of some collaborative partnerships.
(8) A checkerboard assay is being done with the extract and fractions but there is no information provided on what the active element or compound composition of the extract. Seems like a premature assay when you don’t have the active component.
Answer: The checkerboard tests were performed only with the crude methanolic extract and the antifungal amphotericin, fluconazole and ketoconazole. The purpose of the test was the synergism between the drugs for a future indication of combined therapies aimed at reducing the doses of these drugs that already have adverse effects in the literature. In addition, herbal medicines are already a reality and can be used in combination with synthetic medicines.
(9) What is the significance of the propidium iodide marker assay as it relates to identifying mechanism of action.
Answer: PI is a fluorescent dye that interleaves with the DNA and RNA of cells whose plasma membrane integrity has been lost. Therefore, the labeling of cells with this compound allows an analysis of the antifungal activity of OGME in biological membranes, as occurs with the action of amphotericin B.
(10) You demonstrate a dose dependent increase in ROS with your provided methanol fraction that is not related to cell wall or ergosterol. One potential mechanism is the thioredoxin system that functions as a cellular antioxidant. To determine if this pathway is affected by your plant extracts, you can see if glutathione is an antagonist to the inhibitory activity.
Answer: Our group recently published the antifungal activity of thiosemicarbazones against strains of Candida albicans and we verified the protective activity of reduced glutathione against the attack of the molecule in the production of reactive oxygen species. The titration of these two molecules was also titrated by H1 NMR (https://doi.org/10.1016/j.cbi.2020.109028). In this study, these tests were not possible, because the analysis would be subjective, not bringing concrete results because the flavonoid compound is not isolated and identified. Then, we performed the detection test only for reactive oxygen species.
(11) In the abstract, line 25, remove the word “other”.
Answer: The word "other" has been removed from the text.
(12) Please revisit lines 38 and 39 from the abstract. The conclusion statement needs to be revised for English.
Answer: The conclusion of the Abstract section has been revised to improve understanding.
(13) Line 44: “many nutrients” should be changed to “diverse nutrient rich environments...
Answer: Modified text. “Many nutrients” has been changed to “diverse nutrient rich environments”.
(14) Lines 84-86, 99-101, 127, 137-138, 242-244 need editing.
Answer:
Lines 84-86: The text was revised to explain the information regarding the flavonoids found in fraction 5 (FEM5) and OGME.
Lines 99-101: The text has been corrected. The interaction occurs with intracellular proteins.
Line 127: The text has been revised to improve understanding. The crude methanolic extract against Candida krusei presented MIC of 3.12 µg/mL.
Lines 137-138: lines referring to the description of table 02. The text has been modified to explain all the codes used in the table. It was modified to meet the comments of Reviewer 03 regarding all tables in the manuscript.
Lines 242-244: The text has been revised. The description regarding the percentage values of PS released by Candida krusei cells was presented more clearly.
Reviewer 2 Report
The authors of antibiotics-815830 manuscript aimed to characterize Ocotea glomerata extract antifungal activity. Given the increasing resistance rates among Candida spp. searching for new therapeutic options is well justified. However, the manuscript has some serious flaws that require correction. Please see the detailed comments below.
1. The introduction should not start with a sentence on Candida albicans, since the manuscript isn't focused on it. It should rather be a statement about Candida spp. in general.
2. Lines 38-39: Overstatement, given no infection model was studied. Moreover, in lines 99-102 you acknowledge difficulties in using methanol extracts due to their hepatotoxicity.
3. The introduction does not provide sufficient background on current problems with the management of Candida spp. infections e.g. changes in epidemiology (shift towards non-albicans Candida species), drug resistance, etc. that are the reason why we should look for new therapeutic options.
4. Why the components of OGME and fractions weren't identified?
5. Antifungal susceptibility testing
5.A. Extremely limited number of isolates was tested; Isolates antifungal susceptibility background not explained; Species selection not justified.
5.B. Missing controls -> MIC values were not determined for the solvents, e.g. methanol.
5.C. Natural products are known for their batch-to-batch variation. How many batches were analyzed here?
6. Solvent (methanol) impact controls missing in all experiments.
7. Given the poor outcomes of susceptibility testing and associated hepatotoxicity of OGME, what future use do you see for it?
Author Response
(1) The introduction should not start with a sentence on Candida albicans, since the manuscript isn't focused on it. It should rather be a statement about Candida spp. in general.
Answer: Thanks for the comments. The initial text of the Introduction section has been modified to meet the considerations of Reviewer 02.
(2) Lines 38-39: Overstatement, given no infection model was studied. Moreover, in lines 99-102 you acknowledge difficulties in using methanol extracts due to their hepatotoxicity.
Answer: Lines 38-39 referring to the conclusion of the Abstract section indicate the crude extract as promising in the treatment of Candida infections. The literature points to the action of flavonoids against non-albicans Candida species (https://doi.org/10.1007/978-981-10-4732-9_9). Therefore, this conclusion does not go beyond the interpretation of the data obtained in the study.
Lines 99-102 refer to the possibilities of using phenolic compounds to treat fungal infections. However, the hepatotoxicity mentioned in the text concerns those caused by the use of azoles.
(3) The introduction does not provide sufficient background on current problems with the management of Candida spp. infections e.g. changes in epidemiology (shift towards non-albicans Candida species), drug resistance, etc. that are the reason why we should look for new therapeutic options.
Answer: In the Introduction section, information about the current problems with the management of infections by Candida species, in the epidemiological changes and drug resistance was inserted as suggested by Reviewer 02.
(4) Why the components of OGME and fractions weren't identified?
Answer: At the moment we have no collaborations for spectroscopic and spectrometric studies to identify the chemical structure and isolation of the active compound. However, from the fractionation, it was possible to identify the flavonoid class.
(5) Antifungal susceptibility testing
Answer: The section "2.2 Antifungal activity" has been changed to "2.2 Antifungal susceptibility testing" as suggested by Reviewer 02.
(5.A) Extremely limited number of isolates was tested; Isolates antifungal susceptibility background not explained; Species selection not justified.
Answer: The objective of the work was to investigate its antifungal activity against Candida species and to direct the mechanism of action. The extract of Ocotea glomerata was tested for other yeast and filamentous fungi showing good results. However, our group has already consolidated studies for mechanisms of action for Candida species and, therefore, research for filaments will occur later. As for the low number of isolates at work ... our group has sought to analyze the action of the compounds and not just the initial screening of the action spectrum. This is due to the limitations in promoting research in our country (Brazil). Currently, a study with a low number of Candida isolates has been published by our group, analyzing the action mechanism for only one species (https://doi.org/10.1016/j.cbi.2020.109028).
(5.B) Missing controls -> MIC values were not determined for the solvents, e.g. methanol.
Answer: Methanol was not used as a solvent. The methanol described refers to the crude methanolic extract of Ocotea glomerata. These data have already been corrected in the text and tables. The solvent used for the extracts was dimethylsulfoxide (DMSO). Solvent control is not done as it is not foreseen by CLSI. The recommendation is that the DMSO does not exceed 1% in the final test concentrations. This concentration is not able to inhibit fungal cells. Positive control (fungus and RPMI 1640) and negative control (RPMI 1640 only) are performed. In addition, standardization is performed with antifungals and standardized yeasts (ATCC).
(5.C) Natural products are known for their batch-to-batch variation. How many batches were analyzed here?
Answer: In fact, seasonality is an important factor in the production of secondary metabolites. The time referred to the collection of botanical material was inserted in the text. Several collections were carried out between December and February 2015 and, therefore, we considered all experiments carried out with the same batch.
(6) Solvent (methanol) impact controls missing in all experiments.
Answer: Methanol was not used as a solvent. The methanol described in the text refers to the crude methanolic extract of O. glomerata. These data have been corrected in the text and in the tables. The solvent used was dimethylsulfoxide, which did not exceed 1%, as determined by CLSI (See answer 5.B).
(7) Given the poor outcomes of susceptibility testing and associated hepatotoxicity of OGME, what future use do you see for it?
Answer: According to Webster and Collaborators (https://doi.org/10.1016/j.jep.2007.09.014) the results were satisfactory. Hepatotoxicity data refer to azoles such as Fluconazole. The liver damage of the OGME was not mentioned in the text.
Reviewer 3 Report
The manuscript entitled "Ocotea glomerata (Nees) Mez extract and fractions: Chemical characterization, anti-Candida activity and related mechanism of action" presented for review describes the results of laboratory studies aimed at determining antifungal properties of the extract obtained from Ocotea glomerate leaves. Numerous tests have been conducted, but the methodology is ambiguous and needs improvement. The description of results should also be improved. For example, I have not found the results of the antifungal sensitivity test? Please explain this fact. Finally, I don't know if the research covered several species and strains of Candida or maybe C. krusei was the subject of most experiments. I recommend printing the manuscript in a journal Antibiotics (MDPI) after major revision.
Detailed comments:
Table 1-3. Correct in the captions: , on -: for example OGME- O. glomerata methanol extract
L 143-2.3 Checkboard analysis - please enter a description of the method in chapter Material and methods. What was the subject: Species and strains, series, repetitions, control.
L174-192- The authors did not present the results, they wrote ... sorbitol tests showed that OGME does not exert antifungal effect on the cell wall. If the results show nothing, this test can be dispensed with or the results shown despite no differences. This recommendation applies to both Sorbitol and ergosterol tests.
L194-205-B Only C. krusei was tested? What strain? What do the stars in the chart mean? There is no explanation under the graph for the abbreviations introduced by the authors. Abbreviation should be explained under each graph.
L207-217 Only C. krusei was tested? What strain? What do the stars in the chart mean? In the chapter material and methods the Authors claim that they tested Candida spp, while in line 347 there is information that they performed 48 experiments and 10,000 events. Please explain.
L228-256 The methodology states that they tested Candida spp.
L238-239- C. krusei cells exposed to OGME showed an increase in intracellular ROS levels production, proportional to the dosage increase of the compound. Please perform a Pearson's linear correlation analysis.
L265-266. Used as solvents? Please provide specific information on the plant material, GPS position of the site, age of the plants. Was the raw material fresh or dried? Bulk or different variants, repetitions?
L271- The dried extract (150 mg)... Was the test conducted in repetitions, if so please specify in how many?
L284- O. glomerata methanol extract?
L285- Methanol extract samples were .... What samples were tested? Different variants or maybe repetitions?
L297- No description of the experiment in this chapter is presented only for which species and strains were tested. Please give details. The tests were carried out in repetitions? The experiment was carried out in series?
L306- Studies were carried out in two repetitions? Was the series repeated?
L306. I have not found a description of the method before. Please explain.
L306- I couldn't find a description of how the initial inoculum was prepared. Please explain.
L320- The research was carried out in repetitions?
L323- Studies were performed in repetitions? What was on the plates?
L326-330- One series of experiments or more? Repetitions? What was the test material?
L333-What was the subject of the study? What species and strains were tested? How was the experiment designed – positive and negative control?
L342- What Candida were tested? Species? Strains? How was the experiment designed?
L351- What kind of Candida were studied? Species, strains? Please specify the exact conditions of the experiment.
Author Response
(1) Table 1-3. Correct in the captions: , on -: for example OGME- O. glomerata methanol extract
Answer: The modifications to the tables were made following the suggestion of the reviewer 03.
(2) L 143-2.3 Checkboard analysis - please enter a description of the method in chapter Material and methods. What was the subject: Species and strains, series, repetitions, control.
Answer: Section 3.6 of Material and methods contains information on the Checkerboard analysis. Section 3.6 "Synergism test using the broth microdilution technique: checkerboard" was changed to "Checkerboard analysis" as suggested by Reviewer 03. The requested information was also inserted.
(3) L174-192- The authors did not present the results, they wrote ... sorbitol tests showed that OGME does not exert antifungal effect on the cell wall. If the results show nothing, this test can be dispensed with or the results shown despite no differences. This recommendation applies to both Sorbitol and ergosterol tests.
Answer: The ergosterol and sorbitol tests were not presented due to the values being the same as those observed in the MIC. However, the negative result does not exclude the importance of excluding two possible routes of action for the compound.
(4) L194-205-B Only C. krusei was tested? What strain? What do the stars in the chart mean? There is no explanation under the graph for the abbreviations introduced by the authors. Abbreviation should be explained under each graph.
Answer: Only Candida krusei ATCC 6258 was used in the sorbitol and ergosterol tests. This information was inserted in the text. In addition, all abbreviations and symbols used in the figures have been described as suggested by Reviewer 03.
(5) L207-217 Only C. krusei was tested? What strain? What do the stars in the chart mean? In the chapter material and methods the Authors claim that they tested Candida spp, while in line 347 there is information that they performed 48 experiments and 10,000 events. Please explain.
Answer: Only Candida krusei ATCC 6258 was used in the sorbitol and ergosterol tests. This information was inserted in the text. In addition, all abbreviations and symbols used in the figures have been described as suggested by Reviewer 03.
48 experiments were not carried out. The sentence has been corrected. In parallel, the strain (Candida krusei ATCC6258) used in cytometry tests was inserted in the text.
The 10,000 events referred to in the testo is a flow cytometer reading configuration. The equipment performs repeated readings during the cell flow quantifying the detected markings. The determination of these readings (events) are chosen from the literature and recent publications.
(6) L228-256 The methodology states that they tested Candida spp.
Answer: Fixed for Candida krusei ATCC6258.
(7) L238-239- C. krusei cells exposed to OGME showed an increase in intracellular ROS levels production, proportional to the dosage increase of the compound. Please perform a Pearson's linear correlation analysis.
Answer: Pearson's correlation correlates two variables quantitatively. Our data are arranged in percentage values on the X axis. In addition, on the Y axis there is more than one variable: amphotericin, as a standard drug, and the MIC values of the extract. Thus, we will have more than two variables in Y. In this first moment, it was interesting to show the relationship of significance in relation to the control, amphotericin B.
(8) L265-266. Used as solvents? Please provide specific information on the plant material, GPS position of the site, age of the plants. Was the raw material fresh or dried? Bulk or different variants, repetitions?
Answer: The requested information was added in section 3.1. Dry leaves were used after drying in an oven. In addition, data on the collection area and its coordinates were also included.
(9) L271- The dried extract (150 mg)... Was the test conducted in repetitions, if so please specify in how many?
Answer: For the fractionation of the crude methanolic extract 150 mg were used. There were no repetitions and the obtained fractions were directed to antifungal tests again.
(10) L284- O. glomerata methanol extract?
Asnwer: Refers to the crude extract produced with methanol. Crude extracts were also produced with hexane and ethyl acetate. The term has been corrected in the text.
(11) L285- Methanol extract samples were .... What samples were tested? Different variants or maybe repetitions?
Answer: Crude extracts of Ocotea glomerata were produced and tested for their antifungal action. Then, from the results of the antifungal activity, the crude methanolic extract was chosen for fractionation and other biological tests.
(12) L297- No description of the experiment in this chapter is presented only for which species and strains were tested. Please give details. The tests were carried out in repetitions? The experiment was carried out in series?
Answer: The tests were performed according to the standards established by the Clinical and Laboratory Standards Institute (CLSI, 2012) using the M27-S4 protocol. Because the protocol is widely used, it was only cited in the text. However, in section 3.5 the requested information was inserted. In section 2.2, the protocol citation was inserted.
(13) L306- Studies were carried out in two repetitions? Was the series repeated?
Answer: The tests were performed in triplicate with two independent repetitions.
(14) L306. I have not found a description of the method before. Please explain.
Answer: The tests recommended by CLSI are based on microdilutions in broth. Then, starting from serial dilutions of the compounds to be tested, the fungal cells are introduced and, subsequently, incubated and visual reading of the results to obtain the MIC values.
Tests for detection of reactive oxygen species were performed using the CLSI protocol. However, at the end of the test, the wells containing the samples are marked with the probe 2'7'-dichlorodihydrofluorescein (H2DCFDA) and evaluated by flow cytometry in which the fluorescence of the probe is measured when connected to the reactive species. The fluorescence intensity is directly proportional to the amount of reactive oxygen species.
(15) L306- I couldn't find a description of how the initial inoculum was prepared. Please explain.
Answer: A 24-h culture of the tested yeasts was performed on Sabouraud Dextrose Agar (SDA) prepared using an initial inoculum suspension in 5 mL of sterile saline (NaCl, 0.85% saline) where the density was adjusted accordingly to the 0.5 McFarland scale with 90% transmittance determined by spectrophotometry, using a wavelength at 530 nm. This procedure provides a standard yeast concentration containing 1 × 106 to 5 × 106 cells per ml, followed by a 1:100 dilution and then a 1:20 dilution of the standard suspension with RPMI 1640 medium (Roswell Park Memorial Institute Medium) that’s is a growth medium used in cell culture, resulting in a concentration between 5.0 × 102 and 2.5 × 103, where these grew at a temperature of 37 °C.
(16) L320- The research was carried out in repetitions?
Answer: Yes. In triplicate with two independent repetitions.
(17) L323- Studies were performed in repetitions? What was on the plates?
Answer: Yes. In triplicate with two independent repetitions. Flat 96-well plates were used. Triplicates were performed on the same plate and independent repetitions were performed on different days. The dilutions of the compounds, the standard, and the positive and negative control are always inserted in the plates.
(18) L326-330- One series of experiments or more? Repetitions? What was the test material?
Answer: Yes. In triplicate with two independent repetitions. Flat 96-well plates were used. Triplicates were performed on the same plate and independent repetitions were performed on different days. The dilutions of the compounds, the standard, and the positive and negative control are always inserted in the plates.
In the Checkerboard analysis the CLSI protocol is also used. However, serial dilutions occur horizontally and vertically by extracts and antifungals respectively. Thus, cross-dose evaluation is possible.
(19) L333-What was the subject of the study? What species and strains were tested? How was the experiment designed – positive and negative control?
Answer: The ergosterol test was performed with the strain Candida krusei ATCC6258. This information was inserted in section 2.4 as a suggestion from Reviewer 02.
(20) L342- What Candida were tested? Species? Strains? How was the experiment designed?
Answer: The sorbitol test was performed with the strain Candida krusei ATCC6258. This information was inserted in section 2.4 as a suggestion from Reviewer 02.
(21) L351- What kind of Candida were studied? Species, strains? Please specify the exact conditions of the experiment.
Answer: Flow cytometry tests were performed with Candida krusei ATCC6258. Initially, the antifungal susceptibility test was performed following the CLSI protocol. After the incubation period, the cells were marked with propidium iodide and taken to the flow cytometer. The methodology for cytometric analysis is set out in section 3.9. In addition, information about the strain was inserted in section 2.5 as suggested by Reviewer 2.
Round 2
Reviewer 1 Report
Review
- The can still be improved for English language.
- Line 25, remove C. It is non-albicans Candida. Please check throughout paper. Lines 45, 47
- Line 35, “mainly against…”
- Line 35, “complexation mechanisms” is confusing and may not be the word you intend to use here.
- Line 46, correct distribution. I think you mean location.
- Line 54, “pathogens:….”
- Line 56, to many periods after [6].
- Table 1, are the FM5 and OGME fractions that same flavonoids. There are slightly different retention times:25.263 versus 25.377 and 27.177 versus 27.223. Also, this table uses the abbreviations FEM and FM. It is better to select one abbreviation and use consistently.
- Lines 90-91, the statement “in addition to have antiviral, antiproliferative,antimutagenic, antithrombotic and antioxidant activities…” needs to be corrected. “in addition, they present…”
- Line 94, correct “complexation”
- Line 99, 2 periods.
- Line 114, replace “To” with “For”.
- Line 118, replace “have” with “has”. Also, replace “ as” with “as an…”
- “did not evidence an MIC”
- Line 194 and 198: Please change “preventing complexation” and “complexation mechanism”.
- Correct the statement, …” being this one of the first alterations involved in cell death…”.
- Line 325, correct “crude”.
- Line 397, change to “C. krusei”
Author Response
- The can still be improved for English language. THE ENGLISH WAS REVISED
- Line 25, remove C. It is non-albicans Candida. Please check throughout paper. Lines 45, 47 CORRECTED
- Line 35, “mainly against…” CORRECTED
- Line 35, “complexation mechanisms” is confusing and may not be the word you intend to use here. CORRECTED
- Line 46, correct distribution. I think you mean location. CORRECTED
- Line 54, “pathogens:….” CORRECTED
- Line 56, to many periods after [6]. CORRECTED
- Table 1, are the FM5 and OGME fractions that same flavonoids. There are slightly different retention times:25.263 versus 25.377 and 27.177 versus 27.223. DEAR REVIEWER, PROBABLY THERE ARE THE SAME REALLY.
- Also, this table uses the abbreviations FEM and FM. It is better to select one abbreviation and use consistently. CORRECTED
- Lines 90-91, the statement “in addition to have antiviral, antiproliferative,antimutagenic, antithrombotic and antioxidant activities…” needs to be corrected. “in addition, they present…” CORRECTED
- Line 94, correct “complexation” CORRECTED
- Line 99, 2 periods. CORRECTED
- Line 114, replace “To” with “For”. CORRECTED
- Line 118, replace “have” with “has”. Also, replace “ as” with “as an…” CORRECTED
- “did not evidence an MIC” CORRECTED
- Line 194 and 198: Please change “preventing complexation” and “complexation mechanism”. CORRECTED
- Correct the statement, …” being this one of the first alterations involved in cell death…”. CORRECTED
- Line 325, correct “crude”. CORRECTED
- Line 397, change to “C. krusei” CORRECTED
Reviewer 2 Report
The authors responded tolerably to the comments in the first round of review.
MINOR COMMENTS
Lines 45, 47: should be "non-albicans Candida" not "non-Candida albicans"
Table 1: in the table fractions are named FM, whole in the text and legend they are called FEM
Author Response
The authors responded tolerably to the comments in the first round of review.
MINOR COMMENTS
Lines 45, 47: should be "non-albicans Candida" not "non-Candida albicans" CORRECTED
Table 1: in the table fractions are named FM, whole in the text and legend they are called FEM CORRECTED
Reviewer 3 Report
Manuscript entitled "Ocotea glomerata (Nees) Mez extract and fractions: Chemical characterization, anti-Candida activity and related mechanism of action" has been corrected by the authors. In line 268-275 there is a text that should be written in English. After this minor correction I accept the manuscript for printing in Antibiotics JOURNAL.
Author Response
Manuscript entitled "Ocotea glomerata (Nees) Mez extract and fractions: Chemical characterization, anti-Candida activity and related mechanism of action" has been corrected by the authors. In line 268-275 there is a text that should be written in English. After this minor correction I accept the manuscript for printing in Antibiotics JOURNAL.
Answer: Dear reviewer, thank you for your kind comment. The paragraph was rewritten in English.